# Serum Phosphorus as a Risk Factor of Metabolic Syndrome in the Elderly in Taiwan: A Large-Population Cohort Study

**DOI:** 10.3390/nu11102340

**Published:** 2019-10-02

**Authors:** Yi-Han Jhuang, Tung-Wei Kao, Tao-Chun Peng, Wei-Liang Chen, Pi-Kai Chang, Li-Wei Wu

**Affiliations:** 1Department of Surgery, Tri-Service General Hospital, National Defense Medical Center, Taipei 114, Taiwan; jeff22andy@gmail.com (Y.-H.J.); pencil8850@hotmail.com (P.-K.C.); 2Graduate School of Medicine, National Defense Medical Center, Taipei 114, Taiwan; lun0932680266@pchome.com.tw (T.-W.K.); koigogaff@gmail.com (T.-C.P.); weiliang0508@gmail.com (W.-L.C.); 3Division of Family Medicine, Department of Family and Community Medicine, Tri-Service General Hospital, National Defense Medical Center, Taipei 114, Taiwan; 4Division of Geriatric Medicine, Department of Family and Community Medicine, Tri-Service General Hospital, National Defense Medical Center, Taipei 114, Taiwan; 5Division of Colon and Rectal Surgery, Department of Surgery, Tri-Service General Hospital; National Defense Medical Center, Taipei 114, Taiwan

**Keywords:** serum phosphorus, metabolic syndrome, elderly

## Abstract

Background: The impact of serum phosphorus concentration on metabolic syndrome were limited. Therefore, this study aimed to explore the association between the serum phosphorus and incident metabolic syndrome in the elderly in Taiwan. Methods: We included 1491 participants who had health check-ups in the Tri-Service General Hospital for the period 2007 to 2015 and divided them based on age to assess the incidence of metabolic syndrome. We performed the COX regression model to explore the impact of serum phosphorus for metabolic syndrome, diabetes mellitus, and hypertension by an age-specific group. Results: Our result showed that higher serum phosphorus concentration was noted in the elderly in the baseline characteristics. In the group older than 60 years, serum phosphorus concentration was correlated with the incidence of metabolic syndrome (hazard ratios (HR) = 1.39, 95% CI 1.11–1.74) and diabetes mellitus (HR = 1.49, 95% CI 1.15–1.92) after adjustment. We further found the relationship between serum phosphorus and incidence of the components of metabolic syndrome, including higher waist circumference, high-density lipoprotein (HDL), serum triglyceride, and fast glucose. Conclusions: Our study might provide an epidemiological evidence that serum phosphorus was related with the incidence of metabolic syndrome in the elderly in Taiwan.

## 1. Introduction

Serum phosphorus was an essential and important element in the human body, which is absorbed by intestine and reabsorbed by renal proximal tubular. The serum phosphorus concentration is usually determined via three main mechanisms, including dietary intake, gastrointestinal disorder, and renal disorder. Phosphorus metabolism problem often occurred in people who had chronic kidney disease (CKD) and deteriorated aggressively as end stage renal disease (ESRD) [1]. In the previous studies, serum phosphorus was under the regulation of parathyroid hormone (PTH)-vitamin D axis and fibroblast growth factor 23 (FGF23)-bone-kidney axis [2]. FGF23 which is secreted by osteocyte regulated serum phosphorus and vitamin D metabolism. As for overloading dietary phosphorus or increasing 1,25 dihydroxy vitamin D (1,25 D) level, FGF23 is secreted to decrease 1,25 D and PTH concentration [3]. Especially, FGF23 was thought to play an essential role in serum phosphorus balance in patients with CKD [4]. 

Numerous studies have indicated that serum phosphorus was a risk factor in cardiovascular outcome and related mortality [5,6,7]. The mechanisms are endothelial dysfunction [8,9,10], vascular stiffness [11,12], and vascular calcification [13,14]. Furthermore, Vart et al. also demonstrated that people with hypertension and elevated serum phosphorus were at the higher risk of cardiovascular mortality in NHANES-III. In Taiwan, according to the National Health Promotion Administration, the prevalence of metabolic syndrome increased up to 19.7% in grown-up people in 2007. Moreover, cardiovascular disease (CVD), type 2 diabetes mellitus (T2DM), and hypertension (HTN) might be related with metabolic syndrome, which have been a public health issue and the prominent cause of death in Taiwan. Meanwhile, our country has entered the aging society in recent decades. However, researches investigating the association between serum phosphorus and metabolic syndrome had not shown consistent results [15]. The purpose of our study was to investigate the association between serum phosphorus and incident metabolic disease in the elderly in the large cohort study in Taiwan.

## 2. Materials and Methods

### 2.1. Study Subjects

We obtained the data from health check-ups in the Tri-Service General Hospital (TSGH) for the period 2007 to 2015.

The Institutional Review Board (IRB) of Tri-Service General Hospital approved the study and waived the informed consents because we analyzed our data only for medical research and privately. Initially, we included 52,666 participants who received regular health check-ups. Then, the exclusion criteria were validated as the following. First, we eliminated 2604 individuals with previous history of CVD, CKD, HTN, and T2DM. Then, we eliminated 543 individuals who had acute and chronic disease or took any medication that might affect hematological parameters or metabolism. Third, we eliminated 48,028 individuals without complete data of MetS components, laboratory reports, and physical examination. Ultimately, 491 participants were registered for further analysis (Figure 1). To investigate the aging effect on the association between serum phosphorus concentration and metabolic disease, we divided all subjects into three groups: Age less than 39 years old, age between 40 and 59 years old, and age more than 60 years old.

### 2.2. Measurement of Serum Phosphorus

We used the Hitachi model 737 multichannel analyzer (Boehringer Mannheim Diagnostics, Indianapolis, IN, USA) to quantify the concentration of serum phosphorus.

#### 2.2.1. Covariates

Information about medical history, family history, and current prescriptions of participants were collected through the questionnaire. Qualified physicians performed a series of interviews and physical examination. Body mass index (BMI) was calculated as the ratio of body mass tissue (including muscle, fat, and bone) in a unit of kg/m^2^. We measured each individual’s blood pressure three times after participants sat and relaxed for 5 min and recorded the average values. The cuff was usually placed on the participants’ right arm without contraindication.

We collected peripheral blood sampling obtained after fasting for 8 h for serum biochemistry profile, including serum total cholesterol (TC), triglyceride (TG), high and low-density lipoprotein cholesterol (HDL-C; LDL-C), fasting glucose, creatinine, uric acid (UA), phosphorus and high-sensitivity C-reactive protein (hs-CRP). All biochemistry data were analyzed by a series of reactions in the laboratory.

#### 2.2.2. The Definition of Hypertension (HTN), Diabetes Mellitus (DM), and Metabolic Syndrome (MetS)

Individuals with systolic blood pressure/diastolic blood pressure higher than 140/90 mmHg or having antihypertensive agents previously were diagnosed of hypertension [16]. Individuals with fasting serum glucose ≥126 mg/dL, glycated hemoglobin (HbA1c) ≥6.5% or having antidiabetic agents were diagnosed of diabetes mellitus [17]. Metabolic syndrome was defined as the individual with central obesity (base on the ethnicity-specific waist circumference cut-off point, ≥90 cm in men and ≥80 cm in women for Chinese) having at least two of the five following components [18]:(1)Triglycerides ≥150 mg/dL (≥1.7 mmol/L)(2)HDL cholesterol <40 mg/dL (<1.03 mmol/L) in men or <50 mg/dL (<1.29 mmol/L) in women(3)Systolic blood pressure (SBP) ≥130 mm Hg or diastolic blood pressure (DBP) ≥85 mmHg(4)Fasting glucose ≥100 mg/dL (≥5.6 mmol/L) or be diagnosed of T2DM

All potential MetS cases were validated by follow-up questionnaires and diagnosis by physicians, and then classified into the MetS group.

### 2.3. Statistical analysis

We performed IBM SPSS Statistics (IBM Corp. Released 2016. IBM SPSS Statistics for MAC, Version 24.0. IBM Corp., Armonk, NY, USA) for analysis. Continuous variables were described in terms of their mean and standard deviation, while discrete variables were described in terms of their frequency and percentages. The Student’s t-test and the Chi-square test were used to demonstrate differences in demographic characteristics in Table 1. In Table 2, we applied the COX regression model to calculate the hazard ratios (HRs) of serum phosphorus for MetS, T2DM, and HTN between the age-specific group. The model was adjusted for age, sex, BMI, family history, serum total cholesterol, serum creatinine, and serum uric acid level because of significant differences in demographic characteristics by age group, the Covariate adjustment was also done. We further stratified metabolic syndrome into each component according to International Diabetes Federation metabolic syndrome criteria and conducted the adjusted model to assess the further relationship between metabolic syndrome components and phosphorus concentration. Statistical significance was indicated as two-sided p values was less than 0.05.

## 3. Results

### 3.1. Participant Characteristics

There was a collection of baseline characteristics classified by age in Table 1. The older participants who were older than 60 years tended to have higher BMI (28.83 ± 8.15), waist circumference, systolic, and diastolic blood pressure (126.27 ± 20.56; 77.10 ± 11.98), serum glucose (157.62 ± 55.4), serum total cholesterol (192.47 ± 34.25), serum creatinine (0.87 ± 0.23), serum uric acid (5.69 ± 1.34) and serum phosphorus (4.17 ± 1.39), significantly.

### 3.2. Association between the Metabolic Diseases and Serum Phosphorus Level

In Table 2, we investigated the correlation between serum phosphorus level and metabolic diseases. In the group aged younger than 60 years old, there was no significant correlation between serum phosphorus concentration and metabolic diseases. Inversely, in the group older than 60 years, serum phosphorus concentration was associated with incidence of metabolic syndrome (HR = 1.39, 95% CI 1.11–1.74) and diabetes mellitus (HR = 1.49, 95% CI 1.15–1.92), except for hypertension after adjustment. The numbers of metabolic syndrome, diabetes mellitus, and hypertension were shown in Appendix A.

### 3.3. Association between the Metabolic Syndrome Components and Serum Phosphorus Level

Furthermore, we assessed the correlation between incidence of MetS components and serum phosphorus level in Table 3. In COX regression model, hazard ratios of serum phosphorus level for waist circumference, HDL, serum triglyceride, and fast glucose were significant (1.18 (95% CI 1.06–1.31), 1.27 (95% CI 1.07–1.50), 1.41 (95% CI 1.15–1.72) and 1.32 (95% CI 1.14–1.53), separately) in group aged older than 60 years after adjustment.

## 4. Discussions

The increasing incidence of diabetes mellitus has been observed worldwide in recent decades [19]. The HRs of serum phosphorus in the elderly with diabetes mellitus was 1.487 (95% CI: 1.151, 1.922, *p* < 0.05). We demonstrated serum phosphorus concentration to be a risk of diabetes mellitus in the elderly, which was consistent with the previous study [20]. Serum phosphorus concentration might also be associated with fasting sugar concentration in the elderly (Table 3). Furthermore, our results showed the serum phosphorus concentration to be an important risk of MetS and its components, including BMI, serum HDL, serum triglyceride, and fast plasma glucose in the elderly in Taiwan. In previous studies, high level of FGF23 was also linked with insulin-resistance, which was related to metabolic syndrome [21]. FGF23 was associated with regulation of serum phosphorus concentration in individuals with diabetes mellitus [20]. One of the explanatory mechanisms was that serum phosphorus regulated by FGF23 was an important role in energy generation and protein production. Furthermore, FGF19 and FGF21 had been approved to be associated with fat mass and glucose metabolism in animal model [22]. FGF23 was also reported to have similar function of FGF19 and FGF21 in previous study [23]. Consistent with the previous study, FGF23 might be relevant to MetS via systemic phosphorus metabolism [24].

Inconsistent with previous studies, a prospective cohort study demonstrated that the positive joint effect of serum phosphorus and hypertension on cardiovascular mortality, suggest that the serum phosphorus and hypertension shared the same pathogenesis pathway of cardiovascular disease, vessel calcified, and injury [25]. Furthermore, there was significant relationship between higher serum phosphorus concentration and coronary artery calcification [26].

To best of our knowledge, compared with previous cross-sectional studies, this is the cohort study to investigate the relationship between MetS and serum phosphorus. Moreover, a significant association was observed in participants older than 60 years old. However, there were some limitations of our study. First, we used a single method to measure serum phosphorus level. Even with exclusion and blood collection after 8-h fasting, serum phosphorus would be influenced by lots of factors, such as dietary intake, dietary supplement, or diurnal hormonal rhythm. The variation of serum phosphorus might be up to 50% over time [27]. Second, some confounding factors, such as serum parathyroid hormone concentration, serum vitamin D concentration, and fibroblast growth factor-23 were not included in our study. Those confounding factors might help us investigate the serum phosphorus concentration regulation.

## 5. Conclusions

Our study indicated higher serum phosphorus level as an important risk factor of DM and MetS in the elderly in Taiwan. Moreover, the incidence of MetS components (such as waist circumference, HDL, TG, fast glucose) were also associated with serum phosphorus after adjustment. Although further researches are needed in the future, our study might provide an epidemiologic evidence that serum phosphorus was related with the incidence of MetS and DM in the elderly in Taiwan.

## Figures and Tables

**Figure 1 nutrients-11-02340-f001:**
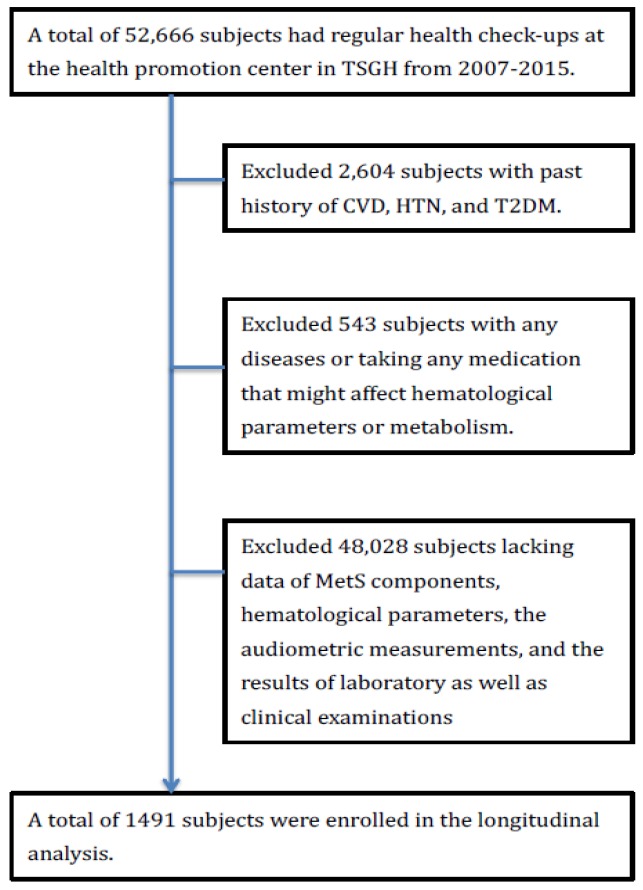
Description of the study flowchart. TSGH, Tri-Service General Hospital, CVD, cardiovascular disease, T2DM, type 2 diabetes mellitus, HTN, hypertension.

**Table 1 nutrients-11-02340-t001:** Baseline demographic data.

	Group 1 Age < 40	Group 2 40 ≤ Age < 60	Group 3 60 ≤ Age
**Continuous Variables**			
Sample size	698	549	244
Age (years)	29.26 (5.57)	49.21 (5.64)	69.21 (8.05)
BMI (kg/m^2^)	22.99 (4.10)	25.05(41.88)	28.83(115.74)
Waist circumference (cm)			
Male	83.95 (10.20)	86.30 (8.95)	87.56 (8.39)
Female	71.36 (10.01)	77.17 (9.53)	81.76 (9.10)
Systolic BP (mmHg)	111.49 (20.95)	115.96 (18.59)	126.27 (20.56)
Diastolic BP (mmHg)	69.54 (10.52)	74.71 (11.84)	77.10 (11.98)
Serum fast glucose	119.46 (33.65)	137.79 (43.63)	157.62 (55.40)
Serum HDL-cholesterol (mg/dL)	58.14 (15.77)	56.85 (16.31)	55.49 (14.80)
Serum triglycerides (mg/dL)	88.92 (68.76)	127.37 (103.23)	122.32 (76.22)
Serum total cholesterol (mg/dL)	174.78 (31.67)	194.83 (34.92)	192.47 (34.25)
Serum LDL-cholesterol (mg/dL)	109.32 (29.04)	125.90 (33.09)	119.17 (31.07)
Serum creatinine (mg/dL)	0.81 (0.28)	0.83 (0.45)	0.87 (0.23)
Serum phosphorus (mg/dL)	3.80 (0.63)	4.01 (2.34)	4.17 (1.389)
Serum uric acid (mg/dL)	5.50 (2.13)	5.52 (1.48)	5.69 (1.34)
Serum Hs-CRP (mg/L)	0.19 (0.30)	0.24 (0.54)	0.24 (0.50)
**Categorical Variables**			
Male, *n* (%)	419 (50)	330 (50)	146 (50)
Family history of CVD, *n* (%)	324 (46.5)	222 (40.6)	31 (12.9)

Demographic data of groups was divided by age in the study and all variables represented as continuous variables (mean and standard deviation) and categorical variables (frequency). Abbreviation: BMI, body mass index; BP, blood pressure; LDL, low-density lipoprotein; HDL, high-density lipoprotein; Hs-CRP, high sensitivity C-reactive protein; CVD, cardiovascular disease.

**Table 2 nutrients-11-02340-t002:** Hazard ratios of serum phosphorus level for metabolic syndrome, diabetes mellitus, and hypertension.

		HR (95% CI)	HR (95% CI)	HR (95% CI)
		Group Age 1–39	Group Age 40–59	Group Age 60–
**Serum phosphorus concentration**		3.80 (0.63)	4.01 (2.34)	4.17 (1.389)
**Model 1**	Metabolic syndrome	0.78 (0.49, 1.26)	1.00 (0.91, 1.09)	1.36 (1.12, 1.65) **
	Diabetes mellitus	0.53 (0.13, 2.17)	1.01 (0.92, 1.11)	1.55 (1.22, 1.96) **
	Hypertension	0.78 (0.51, 1.20)	1.03 (0.99, 1.07)	1.10 (0.90, 1.33)
**Model 2**	Metabolic syndrome	0.82 (0.55, 1.22)	1.00 (0.89, 1.12)	1.34 (1.08, 1.65) **
	Diabetes mellitus	0.58 (0.16, 2.11)	1.01 (0.92, 1.13)	1.50 (1.17, 1.91) **
	Hypertension	0.85 (0.59, 1.22)	1.03 (1.00, 1.07)	1.00 (0.81, 1.23)
**Model 3**	Metabolic syndrome	0.75 (0.49, 1.16)	0.99 (0.85, 1.15)	1.39 (1.11, 1.74) **
	Diabetes mellitus	0.31 (0.06, 1.75)	1.01 (0.91, 1.13)	1.49 (1.15, 1.92) **
	Hypertension	0.83 (0.57, 1.21)	1.04 (1.00, 1.08)	0.96 (0.77, 1.20)

Hazard ratios of serum phosphorus level for metabolic syndrome, diabetes mellitus, and hypertension. Model 1 was unadjusted; age, sex, family history and BMI were adjusted in model 2; serum total cholesterol, serum creatinine, and serum uric acid were further adjusted in model 3. HR, hazard ratios. (** *p* < 0.01).

**Table 3 nutrients-11-02340-t003:** Hazard ratios of serum phosphorus level for the metabolic syndrome components.

		HR (95% CI)	HR (95% CI)	HR (95% CI)
		Group Age 1–39	Group Age 40–59	Group Age 60–
**Serum phosphorus concentration**		3.80 (0.63)	4.01 (2.34)	4.17 (1.389)
**Model 1**	Blood Pressure	0.96 (0.67, 1.39)	0.99 (0.91, 1.08)	1.27 (1.05, 1.54) *
	Waist circumference	0.90 (0.80, 1.01)	1.01 (0.99, 1.04)	1.28 (1.16, 1.41)
	HDL	0.86 (0.66, 1.12)	1.01 (0.97, 1.06)	1.38 (1.18, 1.61) **
	Triglyceride	0.97 (0.69, 1.36)	1.02 (0.98, 1.07)	1.46 (1.22, 1.74) **
	Fast glucose	0.92 (0.60, 1.45)	1.01 (0.95, 1.01)	1.43 (1.25, 1.64) **
**Model 2**	Blood Pressure	1.06 (0.77, 1.46)	0.99 (0.90, 1.09)	1.12 (0.90, 1.38)
	Waist circumference	0.81 (0.71, 0.92) **	1.02 (0.99, 1.04)	1.18 (1.06, 1.31) **
	HDL	0.80 (0.60, 1.06)	1.02 (0.96, 1.07)	1.26 (1.06, 1.48) **
	Triglyceride	1.04 (0.78, 1.38)	1.03 (0.99, 1.07)	1.36 (1.12, 1.65) **
	Fast glucose	0.98 (0.67, 1.44)	1.00 (0.94, 1.08)	1.33 (1.15, 1.54) **
**Model 3**	Blood Pressure	1.06 (0.76, 1.47)	0.99 (0.89, 1.11)	1.09 (0.87, 1.35)
	Waist circumference	0.81 (0.72, 0.93) **	1.01 (0.98, 1.04)	1.18 (1.06, 1.31) **
	HDL	0.78 (0.59, 1.03)	1.02 (0.97, 1.07)	1.27 (1.07, 1.50) **
	Triglyceride	1.03 (0.76, 1.38)	1.03 (0.99, 1.08)	1.41 (1.15, 1.72) **
	Fast glucose	0.96 (0.64, 1.44)	1.00 (0.93, 1.08)	1.32 (1.14, 1.53) **

Hazard ratios of serum phosphorus level for the metabolic syndrome components. Model 1 was unadjusted; age, sex, family history, and BMI were adjusted in model 2; serum total cholesterol, serum creatinine, and serum uric acid were further adjusted in model 3. (* *p* < 0.05, ** *p* < 0.01).

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
