# Peer review of "Serum Phosphorus as a Risk Factor of Metabolic Syndrome in the Elderly in Taiwan: A Large-Population Cohort Study"

_nutrients, 2019, doi:10.3390/nu11102340_

Round 1
Reviewer 1 Report
This manuscript has a major limitation.
1. This manuscript have to show another analized data of serum P concentration by category group in each age-agroup.
2. Table 2,3 showed no serum mean P concentration to get the HR in each age group.
Please show another data results or build up the results.
Author Response
-This manuscript has to show another analyzed data of serum P concentration by category group in each age-group.
Response:
Thank you for your thorough review and salient observations. In table 1, we showed mean serum phosphorus concentration in each age group.
-Table 2,3 showed no serum mean P concentration to get the HR in each age group.
Response:
Thank you for your thorough review and salient observations. We revised our table 2, 3 and added mean serum phosphorus concentration in each age group.
Reviewer 2 Report
The article is very well written. The results are well presented and the conclusions drawn fit nicely. I will be only wanting to there Are there any publicly available data which the authors can look into to correlate serum phosphorous levels on any other population other than Taiwan.
Author Response
- The article is very well written. The results are well presented, and the conclusions was drawn fit nicely. I will be only wanting to there are any publicly available data which the authors can look into to correlated serum phosphorous levels on any other population other than Taiwan.
Response:
Thank you for your thorough review and salient observations. In our research, we only collect data from health check-up exams at the General Health Promotion Center in the Tri-Service General Hospital nationwide and further work-up might be done in the future.
Reviewer 3 Report
There are some very interesting results presented, however they are not presented very well and the results are not completely
The inclusion criteria for this study has to be defined better. Which is the rational for dividing by the age? Statistical analysis was not reported for Table 1. Table 2 is not clear.P lease, explain which models were used for the statistical analysis. Why did they use cholesterol,creatinine and uric acid for Model 2 and 3? The number of subjects with metabolic syndrome, diabetes. and with hypertesion has to be reported. I suggest to divide the subgroup with age >60 in three groups: Diabetes, metabolic. syndrome and. hypertension and to investigate the relationship. with serum phosphorus. Have youj any data about Vitamin D and parathyroid hormone in these patients?
Author Response
- The inclusion criteria for this study has to be defined better. Which is the rational for dividing by the age?
Response:
Thank you for your thorough review and salient observations. In our research, we included individuals who received health check-ups in our hospital and excluded them according to our exclusion criteria. Finally, a total of 1,491 individuals were enrolled. Because we want to explore the age impact on the association between serum phosphorus concentration and metabolic disease as for aging society, we divided our individuals into age group.
- Statistical analysis was not reported for Table 1.
Response:
Thank you for your thorough review and salient observations. According to your recommendation, we revised our “statistical analysis” section. The Student’s t-test and the Chi-square test were used to demonstrate differences in demographic characteristics in table 1. Continuous variables were described in terms of their mean and standard deviation, while discrete variables were described in terms of their frequency and percentages.
-Table 2 is not clear. Please, explain which models were used for the statistical analysis. Why did they use cholesterol, creatinine and uric acid for Model 2 and 3?
Response:
Thank you for your thorough review and salient observations. We applied the COX regression model to investigate the hazard ratios of serum phosphorus for MetS, T2DM and HTN between age-specific group in table 2. We further used serum cholesterol, creatinine and uric acid because of significant difference in demographic characteristics by age group.
-The number of subjects with metabolic syndrome, diabetes and hypertension has to be reported. I suggest dividing the subgroup with age >60 in three groups: diabetes, metabolic syndrome and hypertension and to investigate the relationship with serum phosphorus.
Response:
Thank you for your thorough review and salient observations. We added the number of subjects with metabolic syndrome, diabetes and hypertension as supplementary table S1. In table 2, we showed the association between serum phosphorus and diabetes, metabolic syndrome and hypertension among individual with age >60.
- Have you any data about Vitamin D and parathyroid hormone in these patients?
Response:
Thank you for your thorough review and salient observations. Unfortunately, we did have data about Vitamin D and parathyroid hormone in these individuals because those data were not collected according to routine health check-ups. Therefore, it is the limitation of our study and further work-up might be needed to help us understand the serum phosphorus concentration regulation.
Round 2
Reviewer 1 Report
Thank you for your manuscript revision.
Good revision was made.